# Assessing university guidance and tutoring in higher education: Validating a questionnaire on Ecuadorian students

María Isabel Amor[1], Kasandra Vanessa Saldarriaga Villamil[2‡], Irene Dios[3] *

1 Department of Education, University of Cordoba, Cordoba, Spain, 2 Technical University of Manabi, Portoviejo, Ecuador, 3 Department of Psychology, University of Cordoba, Cordoba, Spain

☯ These authors contributed equally to this work.
‡ This author also contributed equally to this work.
* irene.dios@uco.es

**Data Availability Statement:** All relevant data are within the paper.

**Funding:** The authors received no specific funding for this work.

## Abstract

This study was intended to explore and confirm the factorial structure and to analyze the psychometric properties of an instrument for university guidance and tutoring, apply it, and detect differences between sociodemographic variables. A total of 1,048 students from five universities in the province of Manabi (Ecuador) participated. The study was divided into two phases with differentiated samples. An exploratory phase, made up of 200 subjects (19.1%), and another confirmatory phase, made up of 848 (80.9%), where the questionnaire was also applied. The results supported the three-factor structure of the instrument called "Questionnaire for the Assessment of Guidance and Tutoring in Higher Education" (Q-AGT), with of a total of 21 items. The indices of goodness of fit, reliability and internal consistency of the model were considered satisfactory. The application of the questionnaire did not show statistically significant differences in the assessment of university guidance and tutoring between men and women, with a high value given by both sexes to the importance of tutoring, the demands and the competences of the teaching staff in the university. The differences were mainly found between universities and branches of knowledge. Among the main conclusions, what stands out is the achievement of a valid and reliable instrument to measure the development of guidance and tutoring in Latin American universities. This contributes to the assessment of university guidance and tutoring as a strategy for the integral development of the student- personally, academically and professionally- and as a possible protective factor against academic dropout.

## Introduction

Guidance and tutoring are especially important in university institutions, as it is a strategy that allows individual counseling and guidance to students about their university studies [1–3]. This, in addition to preventing academic failure and student dropout, also benefits the institution itself by reducing dropout rates and increasing the graduation rates established in the plans and programs [4, 5]. Tutoring is related to higher performance and persistence [6–8].

**Competing interests:** The authors have declared that no competing interests exist.

For this reason, many universities consider the development of guidance and counseling strategies for students to be among the most important aims of their training programs, taking other national and international universities as a reference [9–11].

In recent years, various European researchers have taken up the analysis of the quality of guidance and tutoring in universities [12, 13], which has led to an increase in studies focused on the design and validation of reliable instruments of measurement [14–17]. However, although in Latin American countries there is concern with improving the quality of academic tutoring or counseling at the university [18–22], there is a paucity of measurement instruments allowing scientific knowledge in this field to be obtained [1, 3, 19, 22]. For this reason, it is essential to study the assessment of university tutoring in the countries of South America more deeply, and this involves the creation of reliable measurement instruments. Obtaining reliable instruments for university guidance and tutoring in this context would allow scientific evidence of their quality to be obtained, which would facilitate the extraction of strong and weak points from the programs proposed by the different universities. The improvement of the guidance and tutoring system should be seen as a prevention strategy for academic dropout, an essential matter for any university institution to consider.

## University guidance and tutoring: The case of Ecuador

The Higher Education system in Ecuador is committed to measures that improve the educational quality of universities and guarantee access to all citizens. The introduction of new structures and services aimed at promoting the comprehensive development of students and extending their inclusion at all educational levels, has led to an organizational management model in agreement with the current demands of society in this country [23–25]. The Organic Law of Higher Education [26] establishes guidance as a student right and a teaching role for the faculty. This law creates an administrative unit or department of Student Welfare, among whose functions is the vocational and professional orientation of students. Its mission is to promote the comprehensive development of all students, attending to their interests, needs and personal aspirations, all through principles of prevention, continuity and quality of care and services implemented for the purpose.

Similarly, Art.45 the Organic Law of Intercultural Education [27], on the role of the Deputy Director or Vice Rector, determines that among the functions of this office is to implement pedagogical support and academic tutorials for students according to their needs. Furthermore, the Regulation of Career Path and Ranking of Lecturers and Researchers in the Higher Education System [28] endorses the existence of teacher-tutor guides to accompany students in their academic activities. The purpose is to provide personalized support for the effective use of the opportunities and services that the university offers students, attending to their personal needs, which increases the likelihood of remaining in the institution [22, 29].

However, although education law and the aforementioned regulations have favored an increase in the quality of the Universities of Ecuador, according to the technical report issued by the National Council for the Evaluation and Accreditation of University [30], there are important limitations in a number of areas, among which are the low rate of graduates and the lack of teacher training. Based on the results of this report, the universities began a continuous process of assessment and supervision supported by the Council for the Evaluation, Accreditation and Quality Assurance for Higher Education, to guarantee educational quality and improve the services and infrastructures created for this purpose. In 2013, this council issued the first assessment report [31, 32] and the results obtained in relation to the criterion of academic efficiency, in which the effective strategies for admission, retention and accompaniment

in the educational process of students [32], were that most of the universities in category A and B in Ecuador have a fairly high level.

However, there are still very important limitations [30], related to the insufficient professional training of teachers and the small number of them who have postgraduate studies. Research studies in Latin America [19, 24, 33–35] show that one of the reasons for this situation is that the professional work of the university lecturer is mostly made up of teaching. That is, "teaching classes" is established as a priority activity, and didactic and pedagogical training, and their professional advancement, are of secondary importance. According to Moscoso and Hernández [35], in the university institutions of Ecuador, less importance is given to such training, compared to disciplinary training.

Institutions are currently taking steps to improve the results reported on the low rate of graduate students [30]. Tutoring here becomes importance in Ibero-American universities, since it is considered essential for providing personalized attention to the student, guiding them in their personal, academic and professional development, and as a prevention strategy against academic dropout [20]. However, to be effective, tutoring must be linked to the teaching-learning process [11, 22] acting as an anchoring link between the teaching function and the overall learning of students. To respond to these needs [36], the role of the teaching staff is defined as that of tutor and advisor, who assumes responsibilities beyond teaching practice.

## Scales for university guidance and tutoring

The scientific literature includes many studies on the design and validation of scales, addressing different aspects and dimensions of university orientation and tutoring [16, 37–39]. Most of these studies assess the practice of university tutoring from the perspective of those involved-students and/or teaching staff- [24, 36, 40], by carrying out an extensive review of the management structures and systems used by university institutions for their promotion or improvement.

Some studies focusing on the design and validation of scales consider different dimensions of tutoring in the university setting. Despite the efforts to provide reliable measurement scales for university guidance and tutoring, there are many discrepancies to be considered when addressing the concept [16, 17, 36, 41].

Pérez-Cusó et al. [36] describe and check two subscales for measuring satisfaction with university tutoring: 1) satisfaction with the tutor; and 2) satisfaction with the organization and content. These same authors [1] subsequently set out a scale to assess the needs of university guidance and tutoring, comprising five factors: a) adaptation to the university context; b) identity; c) integration and interpersonal development; d) teaching-learning process; and d) professional development. Other authors who contributed a five-factor scale were López-Castro and Pantoja-Vallejo [17], using the following dimensions: 1) functions and tasks of the tutorial activity; 2) satisfaction with the tutorial action in the center; 3) difficulties of tutorial practice; 4) use of technologies in the development of tutorial activities; and 5) knowledge and satisfaction with the work of the counselor.

León-Carrascosa and Fernández-Díaz [16], on the other hand, identify four factorial dimensions for guidance and tutoring in universities: 1) functions of the tutor with the students; 2) functions of the tutor with the families; 3) tutorials; and 4) assessment of tutorials. Another validation study looking at guidance and tutorial practice in universities [13], also opted for a four-factor structure: 1) academic orientation; 2) personal orientation; 3) career guidance; and 4) guidance and ICT.

Among the studies that describe assessment of tutoring with a three-factor model, the study carried out by Delgado-García et al. [15] stands out, proposing the following areas for consideration: 1) functions of the tutor (e.g.: reporting on institutional academic issues); 2) profile of

the tutor (e.g.: affectivity, empathy, attitude. . .); and 3) student needs (e.g.: transition to university).

## The current study

The literature gives great importance to university guidance and tutoring as a strategy in institutions for the development of students, not only academically, but also personally and professionally, as well as offering protection against academic dropout. This leads to the need to evaluate whether this process is carried out properly, with a n increasing amount of research aimed at producing valid and reliable instruments [15, 16, 36, 41, 42]. However, the discrepancies found between studies in relation to their factorial dimensions, and the scarce literature on this type of research in Latin America, require more studies of this kind. For all these reasons, a study to validate an instrument on university guidance and tutoring in Ecuador is here proposed, with the following objectives:

1. To explore and confirm the factorial structure of an instrument on university guidance and tutoring.

2. To analyze the psychometric properties of the instrument.

3. To apply the instrument to identify differences between sexes, universities and university courses.

    Regarding the objectives, the following hypotheses were addressed:

1. The instrument will possess a multidimensional factorial structure.

2. The instrument will have optimal psychometric properties and positive correlation between dimensions.

3. The student body will show similar scores regardless of gender, universities and university degrees.

## Materials and methods

### Participants

1,048 students, from five universities in the province of Manabi, Ecuador, took part in the study, 27.9% from the Technical University of Manabi, 25.9% from the Lay University Eloy Alfaro of Manabi, 10.7% from the State University of the South of Manabi, 9.4% from the University San Gregorio of Portoviejo and 25.9% from the Agricultural Polytechnic School of Manabi. The total sample was chosen by an 'catch-the-eye' procedure, for reasons of accessibility and was divided into two subsamples. The first subsample, comprising, 200 students (19.1%) was used for the exploratory study, and the second, with 848 students (80.9%), for the confirmatory study. Of the total, 44.2% were men and 55.8% women, with ages ranging from 18–41 years (M = 23.73; SD = 3.09).

### Instruments

An ad hoc questionnaire was developed, called "Questionnaire for the Assessment of Guidance and Tutoring in Higher Education" (Q-AGT), made up of a set of questions related to sociodemographic data (e.g.: sex, age, degree, level, university), and a set of 21 items related to university tutoring. Each of the items was assessed by the students using a Likert-type scale with five response options that ranged from 0 ("completely disagree") to 4 ("completely agree"). The items related to the instrument were grouped into 3 subcomponents of tutoring: a) 5 items on

the importance of university tutoring; b) 9 items on the demands of university tutoring; and c) 7 competencies of the teaching staff for university tutoring.

## Procedure and data analysis

Data collection proceeded by contacting each of the university institutions, to inform them and to obtain approval for participation in the research. Once approval was obtained, the questionnaires were completed, which took approximately 30 minutes. The participants, all of legal age, provided written informed consent, were informed about the voluntary nature of the questionnaire, respect for anonymity and the confidentiality of the data provided. All doubts were addressed and data collection proceeded without incident. The process was conducted in accordance with the ethical principles of the Declaration of Helsinki and was approved by the Bioethics Committee of the Technical University of Manabi.

The procedure developed to perform the study involves two different phases. The first phase was exploratory in nature, and examined the factorial structure of the instrument on university tutoring. Specifically, the Factor 10.9.02 program [43] was used to carry out an Exploratory Factor Analysis (EFA) to determine the number of underlying factors. Since the data were ordinal, it was decided to generate a polychoric correlation matrix [44] with the ULS (Unweighted least squares) extraction method. Following the recommendations of various authors [45, 46], Promin oblique rotation was used, as it is adapted to the exploratory nature of the study. For the item-factor saturations, values greater than 0.40 were used as criteria [46, 47].

In the second, confirmatory, phase of the study, the factorial structure of the instrument examined in the preliminary phase was validated, its psychometric properties analyzed and it was then applied. This was done via a Confirmatory Factor Analysis (CFA) using the EQS 6.2 program. In line with the previous phase, on detecting absence of normality, the robust estimation method was used [48]. The fit of the model was interpreted using the Satorra-Bentler chi squared method ($x^2$S-B) and $x^2$S-B/df, considering $\leq 3$ to be optimal, and $\leq 5$ acceptable. Other indices not affected by the sample size were also considered, such as the NNFI; NFI; CFI; IFI, taking values $\leq 0.95$ as criteria for assuming a good fit [49]. Finally, values of RMSEA between 0.05 and 0.08 were also considered to indicate good model fit [50].

Likewise, in this second phase, descriptive and comparative analyzes were carried out with the SPSS 20 program. First, Student's t-test was used to determine the existence of differences between the sex of the participants and the factors found in the scale related to college tutoring. The ANOVA test was also used to verify the existence of differences between the students of the different universities and of the different university degrees with regard to their assessment of university tutoring. To determine among which groups there were differences to be found, Tukey-b or Games-Howell statistics were used, depending on the Levene test. Finally, the effect size (Cohen's d) was evaluated in the t test, considering scores $< 0.5$ as small, between 0.5 and 0.8 as moderate, and $> 0.8$ as large [51]. The confidence level was 95% (p <0.05) and 99% (p <0.01).

## Results

### Exploratory factor analysis of Q-AGT

In the results obtained from the EFA, the Kaiser-Meyer-Olkin test (KMO = 0.78) and the Bartlett's Sphericity Test (Bartlett's Test = 2131.8; p <0.01) indicated that the sample was suitable for carrying out the CFA. The indices on simplicity −index S− and simplicity of loading −index LS− [52, 53] indicated simplicity of factors and that the items were exclusively related to one factor. Table 1 shows the descriptions of each item, as well as the factorial weights and the factor to which each item belongs. The three factors found were called: a) F1 or Importance (IMP); b) F2 or Demand (DEM); and c) F3 or Competence (COM).

**Table 1. Univariant descriptive analysis, factorial loads and AFE.**

| N° | Item | M | SD | Asym. | Curto. | F1 | F2 | F3 | Com. |
|---|---|---|---|---|---|---|---|---|---|
| IMP1 | Provides me with information about the organization and structure of the center in addition to the curriculum | 3.08 | 0.930 | -1.177 | 1.599 | 0.614 | | | 0.574 |
| IMP2 | Assists me in adapting to and integrating with the faculty and university | 3.08 | 0.923 | -1.102 | 1.219 | 0.762 | | | 0.779 |
| IMP3 | Helps me in my academic development | 3.38 | 0.826 | -1.384 | 1.897 | 0.796 | | | 0.544 |
| IMP4 | Guides me in my professional career (professional development) | 3.38 | 0.804 | -1.619 | 3.030 | 0.816 | | | 0.625 |
| IMP5 | Promotes my personal development (promotes autonomy, self-esteem and identity) | 3.42 | 0.956 | -1.510 | 2.259 | 0.755 | | | 0.661 |
| DEM1 | Information | 3.21 | 0.733 | -1.414 | 3.322 | | 0.758 | | 0.561 |
| DEM2 | Academic monitoring | 3.37 | 0.771 | -.968 | 0.962 | | 0.639 | | 0.532 |
| DEM3 | Guidance in my professional career | 3.32 | 0.745 | -1.607 | 3.853 | | 0.636 | | 0.565 |
| DEM4 | Guidance with job placement | 3.42 | 0.828 | -1.339 | 1.728 | | 0.556 | | 0.420 |
| DEM5 | Personal orientation | 3.35 | 0.944 | -1.381 | 2.064 | | 0.812 | | 0.691 |
| DEM6 | Troubleshooting and difficulties | 3.18 | 0.903 | -1.361 | 2.114 | | 0.896 | | 0.733 |
| DEM7 | Decision making | 3.22 | 0.949 | -1.239 | 1.556 | | 0.947 | | 0.706 |
| DEM8 | Helps with level transitions | 3.14 | 0.972 | -1.342 | 1.712 | | 0.943 | | 0.787 |
| DEM9 | Attention to students with disabilities and special needs | 3.15 | 0.840 | -1.520 | 2.568 | | 0.658 | | 0.507 |
| COM1 | General knowledge about college tutoring | 3.38 | 0.719 | -1.505 | 3.531 | | | 0.602 | 0.451 |
| COM2 | Knowledge about the structure and organization of the degree course, as well as the University in general (services, scholarships, activities. . .) | 3.438 | 0.704 | -1.009 | 0.580 | | | 0.765 | 0.643 |
| COM3 | Knowledge about the social and work possibilities of your degree | 3.44 | 0.648 | -1.051 | 1.020 | | | 0.898 | 0.728 |
| COM4 | Knowledge of tutoring techniques (interviews, questionnaires. . .) | 3.44 | 0.782 | -1.544 | 3.088 | | | 0.797 | 0.630 |
| COM5 | Personal characteristics (empathetic, patient, decisive, cordial, mediator, constructive.. . .) | 3.37 | 0.760 | -1.149 | 1.352 | | | 0.784 | 0.696 |
| COM6 | Good intra- and interpersonal relationships | 3.39 | 0.762 | -1.149 | 0.824 | | | 0.783 | 0.626 |
| COM7 | Knows how to give and accept criticism | 3.48 | 0.733 | -1.364 | 1.398 | | | 0.838 | 0.665 |

Note: IMP = Importance; DEM = Demand; COM = Competence.

## Confirmatory factor analysis of Q-AGT

In the second phase, the robust maximum likelihood estimation method was used, due to the lack of normality of the data (Mardia coefficient = 318.9760).

The various goodness of fit indices of the model were considered optimal: $x2S-B$ (186) = 624.3760; p = 0.00 [50]. Those that evaluate the relative fit of the model and are not affected by the sample size, also returned high values, indicating a good fit to the model: NFI = 0.982; NNFI = 0.986; CFI = 0.987; IFI = 0.987; RMSEA = 0.053.

Scores in item-factor correlations ranged from 0.72 (item DEM1) to 0.86 (item COM1). The results revealed a positive correlation between factors (Fig 1), of 0.680 between F1 and F2, 0.555 between F1 and F3, and 0.587 between F2 and F3.

Polychoric correlations between items of the scale were positive (Table 2), ranging from 0.328 y 0.766.

The indices of reliability and internal consistency obtained are considered satisfactory for the instrument in general (α = 0.930) and for each of its factors: a) F1 or Importance (IMP), α = 0.853; b) F2 or Demand (DEM), α = 0.895; and c) F3 or Competence (COM), α = 0.894.

## Application of Q-AGT: Assessment of university tutoring based on sex, university and university degree course

Table 3 shows the participants' sociodemographic data in relation to the 21 items that make up the Q-AGT.

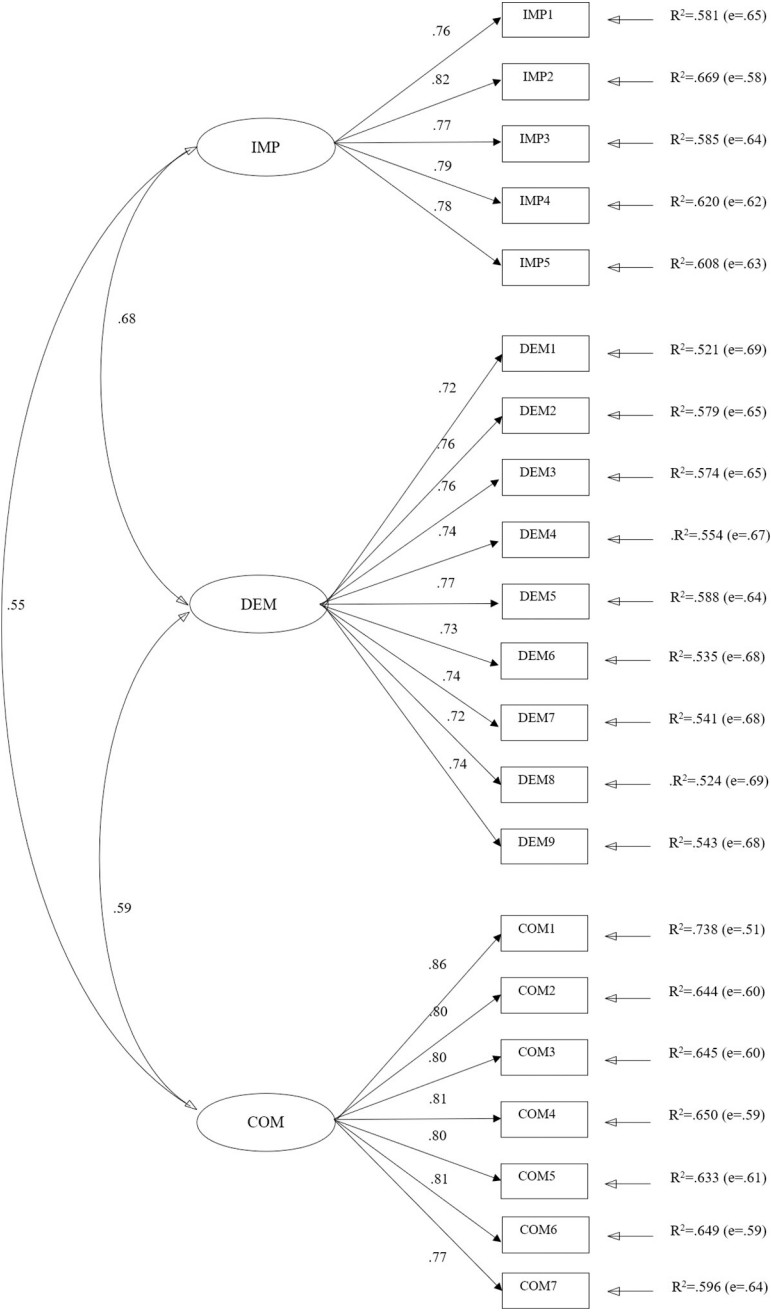

**Fig 1. AFC model for Q-AGT.**

As seen in Table 4, the results did not show statistically significant differences in the assessment of university tutoring between men and women. Both gave a high valuation of the importance of tutoring in the universities of Ecuador, the demands that students make on it and the skills of the teaching staff in undertaking it, related to guidance and counseling.

Regarding the different participating universities, statistically significant differences were detected between universities in the three factors: importance, demand and teaching skills. Table 5 shows the results obtained from the post hoc test, where it is mainly the students of the

**Table 2. Polychoric correlation matrix for items of Q-AGT.**

| | IMP1 | IMP2 | IMP3 | IMP4 | IMP5 | DEM1 | DEM2 | DEM3 | DEM4 | DEM5 | DEM6 | DEM7 | DEM8 | DEM9 | COM1 | COM2 | COM3 | COM4 | COM5 | COM6 | COM7 |
|---|---|---|---|---|---|---|---|---|---|---|---|---|---|---|---|---|---|---|---|---|---|
| IMP1 | 1 | | | | | | | | | | | | | | | | | | | | |
| IMP2 | 0.722 | 1 | | | | | | | | | | | | | | | | | | | |
| IMP3 | 0.538 | 0.658 | 1 | | | | | | | | | | | | | | | | | | |
| IMP4 | 0.567 | 0.622 | 0.766 | 1 | | | | | | | | | | | | | | | | | |
| IMP5 | 0.580 | 0.655 | 0.641 | 0.729 | 1 | | | | | | | | | | | | | | | | |
| DEM1 | 0.495 | 0.494 | 0.416 | 0.435 | 0.471 | 1 | | | | | | | | | | | | | | | |
| DEM2 | 0.475 | 0.518 | 0.492 | 0.471 | 0.469 | 0.636 | 1 | | | | | | | | | | | | | | |
| DEM3 | 0.439 | 0.477 | 0.502 | 0.529 | 0.495 | 0.580 | 0.655 | 1 | | | | | | | | | | | | | |
| DEM4 | 0.420 | 0.463 | 0.409 | 0.488 | 0.492 | 0.540 | 0.610 | 0.722 | 1 | | | | | | | | | | | | |
| DEM5 | 0.451 | 0.482 | 0.424 | 0.504 | 0.565 | 0.540 | 0.609 | 0.613 | 0.665 | 1 | | | | | | | | | | | |
| DEM6 | 0.444 | 0.464 | 0.446 | 0.480 | 0.498 | 0.493 | 0.561 | 0.566 | 0.665 | 0.671 | 1 | | | | | | | | | | |
| DEM7 | 0.406 | 0.475 | 0.398 | 0.448 | 0.537 | 0.483 | 0.553 | 0.523 | 0.545 | 0.696 | 0.650 | 1 | | | | | | | | | |
| DEM8 | 0.399 | 0.447 | 0.360 | 0.405 | 0.539 | 0.451 | 0.544 | 0.533 | 0.545 | 0.634 | 0.645 | 0.710 | 1 | | | | | | | | |
| DEM9 | 0.455 | 0.475 | 0.454 | 0.405 | 0.565 | 0.553 | 0.532 | 0.553 | 0.532 | 0.588 | 0.578 | 0.612 | 0.680 | 1 | | | | | | | |
| COM1 | 0.489 | 0.472 | 0.447 | 0.464 | 0.416 | 0.527 | 0.477 | 0.429 | 0.426 | 0.384 | 0.344 | 0.354 | 0.680 | 0.466 | 1 | | | | | | |
| COM2 | 0.473 | 0.419 | 0.413 | 0.385 | 0.387 | 0.527 | 0.452 | 0.440 | 0.415 | 0.386 | 0.324 | 0.356 | 0.308 | 0.459 | 0.728 | 1 | | | | | |
| COM3 | 0.371 | 0.399 | 0.420 | 0.433 | 0.396 | 0.388 | 0.458 | 0.450 | 0.520 | 0.392 | 0.347 | 0.359 | 0.314 | 0.472 | 0.704 | 0.731 | 1 | | | | |
| COM4 | 0.429 | 0.409 | 0.398 | 0.408 | 0.411 | 0.459 | 0.465 | 0.463 | 0.427 | 0.431 | 0.375 | 0.359 | 0.362 | 0.481 | 0.695 | 0.690 | 0.728 | 1 | | | |
| COM5 | 0.328 | 0.337 | 0.381 | 0.376 | 0.365 | 0.386 | 0.400 | 0.425 | 0.381 | 0.393 | 0.306 | 0.363 | 0.356 | 0.502 | 0.686 | 0.600 | 0.650 | 0.693 | 1 | | |
| COM6 | 0.404 | 0.407 | 0.428 | 0.458 | 0.408 | 0.386 | 0.410 | 0.435 | 0.408 | 0.445 | 0.410 | 0.410 | 0.430 | 0.533 | 0.682 | 0.659 | 0.629 | 0.695 | 0.747 | 1 | |
| COM7 | 0.340 | 0.383 | 0.458 | 0.423 | 0.427 | 0.387 | 0.433 | 0.405 | 0.363 | 0.344 | 0.369 | 0.367 | 0.348 | 0.516 | 0.664 | 0.606 | 0.626 | 0.604 | 0.747 | 0.730 | 1 |

Note: IMP = Importance; DEM = Demand; COM = Competence.

Technical University of Manabi who give a lower valuation of the importance of university tutoring and the demands with respect to other universities, and the same is true of the State University of the South of Manabi, with respect to the Agricultural Polytechnic School of Manabi. With regard to the factor on teaching skill of the tutors, it is the Lay University Eloy Alfaro of Manabi and the State University of the South of Manabi, which return a lower valuation as compared to the Agricultural Polytechnic School of Manabi.

Taking into account the university degrees of the participants, the students in the branch of sciences expressed a greater appreciation of the importance of university tutoring (see Table 6). However, no differences were detected between students of the Science and Social Sciences degrees in the factors related to the demands of tutorials and the skill at university tutoring of the teaching staff.

## Discussion

The first two objectives of this study were to explore and confirm the factorial dimensions of an instrument that we call "Questionnaire for the Assessment of Guidance and Tutoring in Higher Education" (Q-AGT), based on the review of the scientific literature on scales that evaluate university guidance and tutoring [13, 24, 36, 40]. In agreement with other studies, an analysis of the psychometric properties of the instrument was carried out to evaluate the state of tutoring in Latin American universities [16, 37, 38], in order to respond to the lack of research into this matter in Ecuador.

In accordance with previous work [16, 17, 36, 41], the multidimensional nature of the assessment of university tutoring is confirmed. Specifically, and according to other research [15], the instrument was defined by three factors. Factor 1 -Importance- addresses the importance attributed by students to information, organization of the center and the curriculum;

**Table 3. Differences in the scale items for university tutoring based by sex, university, and university degree course.**

| Item N° | Sex | | | | University | | | | | | | | | | University degree courses | | | |
|---|---|---|---|---|---|---|---|---|---|---|---|---|---|---|---|---|---|---|
| | Male | | Female | | UTM | | ULEAM | | USGP | | ESPAM | | UNESUM | | S | | SC | |
| | M | SD | M | SD | M | SD | M | SD | M | SD | M | SD | M | SD | M | SD | M | SD |
| IMP1 | 3.03 | 0.841 | 3.03 | 0.908 | 2.85 | 0.879 | 3.07 | 0.865 | 3.19 | 0.813 | 3.23 | 0.754 | 2.81 | 1.095 | 2.96 | 0.917 | 3.08 | 0.855 |
| IMP2 | 3.09 | 0.824 | 3.11 | 0.852 | 2.84 | 0.884 | 3.14 | 0.812 | 3.29 | 0.697 | 3.3 | 0.723 | 3.04 | 0.977 | 3.02 | 0.866 | 3.16 | 0.819 |
| IMP3 | 3.35 | 0.789 | 3.41 | 0.748 | 3.16 | 0.858 | 3.48 | 0.712 | 3.38 | 0.718 | 3.59 | 0.57 | 3.26 | 0.917 | 3.39 | 0.751 | 3.38 | 0.782 |
| IMP4 | 3.37 | 0.795 | 3.39 | 0.765 | 3.22 | 0.877 | 3.44 | 0.759 | 3.35 | 0.748 | 3.51 | 0.652 | 3.4 | 0.801 | 3.4 | 0.776 | 3.36 | 0.786 |
| IMP5 | 3.22 | 0.843 | 3.21 | 0.919 | 2.93 | 1.037 | 3.3 | 0.829 | 3.36 | 0.733 | 3.45 | 0.678 | 3.04 | 0.942 | 3.22 | 0.879 | 3.21 | 0.9 |
| DEM1 | 3.38 | 0.769 | 3.35 | 0.833 | 3.24 | 0.922 | 3.36 | 0.766 | 3.38 | 0.718 | 3.5 | 0.616 | 3.32 | 0.987 | 3.33 | 0.825 | 3.38 | 0.797 |
| DEM2 | 3.27 | 0.815 | 3.30 | 0.777 | 3.13 | 0.869 | 3.32 | 0.733 | 3.4 | 0.608 | 3.46 | 0.711 | 3.13 | 0.95 | 3.19 | 0.832 | 3.36 | 0.761 |
| DEM3 | 3.33 | 0.809 | 3.39 | 0.780 | 3.25 | 0.846 | 3.39 | 0.794 | 3.4 | 0.686 | 3.48 | 0.737 | 3.3 | 0.837 | 3.31 | 0.852 | 3.4 | 0.755 |
| DEM4 | 3.26 | 0.862 | 3.30 | 0.821 | 3.24 | 0.885 | 3.26 | 0.819 | 3.4 | 0.722 | 3.32 | 0.828 | 3.23 | 0.883 | 3.18 | 0.902 | 3.34 | 0.794 |
| DEM5 | 3.18 | 0.903 | 3.15 | 0.922 | 2.95 | 1.024 | 3.15 | 0.851 | 3.38 | 0.718 | 3.34 | 0.822 | 3.15 | 0.999 | 3.11 | 0.94 | 3.19 | 0.899 |
| DEM6 | 3.26 | 0.832 | 3.22 | 0.876 | 3.02 | 0.976 | 3.22 | 0.826 | 3.4 | 0.722 | 3.43 | 0.734 | 3.23 | 0.857 | 3.23 | 0.875 | 3.25 | 0.843 |
| DEM7 | 3.15 | 0.919 | 3.11 | 0.973 | 2.86 | 1.123 | 3.12 | 0.885 | 3.36 | 0.698 | 3.4 | 0.79 | 3 | 0.943 | 3.08 | 0.924 | 3.16 | 0.969 |
| DEM8 | 3.19 | 0.911 | 3.07 | 0.991 | 2.83 | 1.167 | 3.12 | 0.87 | 3.33 | 0.708 | 3.36 | 0.819 | 3.14 | 0.868 | 3.07 | 0.958 | 3.16 | 0.959 |
| DEM9 | 3.37 | 0.904 | 3.33 | 0.899 | 3.24 | 1.063 | 3.28 | 0.9 | 3.45 | 0.673 | 3.54 | 0.73 | 3.25 | 0.926 | 3.29 | 0.961 | 3.39 | 0.865 |
| COM1 | 3.41 | 0.768 | 3.45 | 0.708 | 3.37 | 0.774 | 3.44 | 0.671 | 3.48 | 0.675 | 3.55 | 0.636 | 3.29 | 0.986 | 3.4 | 0.788 | 3.47 | 0.683 |
| COM2 | 3.37 | 0.724 | 3.43 | 0.697 | 3.43 | 0.695 | 3.3 | 0.741 | 3.53 | 0.616 | 3.5 | 0.637 | 3.23 | 0.844 | 3.3 | 0.776 | 3.48 | 0.641 |
| COM3 | 3.38 | 0.723 | 3.42 | 0.705 | 3.46 | 0.722 | 3.31 | 0.751 | 3.4 | 0.756 | 3.48 | 0.6 | 3.3 | 0.782 | 3.33 | 0.786 | 3.47 | 0.643 |
| COM4 | 3.34 | 0.755 | 3.42 | 0.741 | 3.37 | 0.789 | 3.3 | 0.799 | 3.39 | 0.646 | 3.54 | 0.614 | 3.29 | 0.838 | 3.33 | 0.781 | 3.43 | 0.709 |
| COM5 | 3.35 | 0.799 | 3.41 | 0.769 | 3.43 | 0.765 | 3.23 | 0.908 | 3.41 | 0.65 | 3.51 | 0.665 | 3.29 | 0.807 | 3.32 | 0.865 | 3.43 | 0.717 |
| COM6 | 3.35 | 0.776 | 3.44 | 0.730 | 3.41 | 0.735 | 3.31 | 0.82 | 3.38 | 0.663 | 3.53 | 0.651 | 3.31 | 0.878 | 3.32 | 0.819 | 3.46 | 0.7 |
| COM7 | 3.45 | 0.760 | 3.51 | 0.685 | 3.49 | 0.686 | 3.41 | 0.825 | 3.39 | 0.646 | 3.61 | 0.628 | 3.44 | 0.763 | 3.48 | 0.747 | 3.48 | 0.698 |

Note: IMP = Importance; DEM = Demand; COM = Competence; UTM = Technical University of Manabi; ULEAM = Lay University Eloy Alfaro of Manabi;
USGP = University San Gregorio of Portoviejo; UNESUM = State University of the South of Manabi; ESPAM = Agricultural Polytechnic School of Manabi;
S = Sciences; SC = Social Sciences.

adaptation and integration with the university; help with academic, personal and professional development. The second factor -Demand-, assesses the levels expressed by students about the need for information; monitoring academic, personal and professional guidance; problem solving and difficulties; decision making; help in level transitions; and attention to diversity. Finally, the third factor -Competence- addresses the teaching competence that, from the student perspective, is necessary for the proper practice of university guidance and tutoring, such as competence in university tutoring and organization; competence in social insertion;

**Table 4. Differences in the scale factors for university tutoring based by sex.**

| Factor | Sex | | | | t student | | |
|---|---|---|---|---|---|---|---|
| | Male (n = 372) | | Female (n = 476) | | | | |
| | M | SD | M | SD | t | p | d |
| IMP | 3.21 | 0.657 | 3.23 | 0.662 | 0.417 | 0.677 ns | 0.10 |
| DEM | 3.26 | 0.635 | 3.25 | 0.660 | 0.371 | 0.711 ns | 0.05 |
| COM | 3.38 | 0.556 | 3.44 | 0.563 | -1.513 | 0.131 ns | 0.15 |

Note: p = significance; ns = not significant; d = Cohen's d; IMP = Importance; DEM = Demand;
COM = Competence.

**Table 5. Differences in the scale factors for university tutoring by university.**

| Factor | University | n | M | SD | ANOVA | | |
|---|---|---|---|---|---|---|---|
| | | | | | $F_{(4, 843)}$ | p | Differences between groups |
| **IMP** | UTM | 237 | 3.00 | 0.698 | 13.901 | 0.00** | UTM<ULEAM |
| | | | | | | | UTM<USGP |
| | ULEAM | 220 | 3.29 | 0.624 | | | UTM<ESPAM |
| | USGP | 80 | 3.31 | 0.618 | | | UNESUM<ESPAM |
| | ESPAM | 220 | 3.42 | 0.530 | | | |
| | UNESUM | 91 | 3.11 | 0.772 | | | |
| **DEM** | UTM | 237 | 3.08 | 0.730 | 9.696 | 0.00** | UTM<USGP |
| | | | | | | | UTM<ESPAM |
| | ULEAM | 220 | 3.25 | 0.556 | | | ULEAM<ESPAM |
| | USGP | 80 | 3.39 | 0.547 | | | UNESUM<ESPAM |
| | ESPAM | 220 | 3.43 | 0.559 | | | |
| | UNESUM | 91 | 3.20 | 0.711 | | | |
| **COM** | UTM | 237 | 3.42 | 0.594 | 4.451 | 0.01** | ULEAM<ESPAM |
| | ULEAM | 220 | 3.33 | 0.578 | | | |
| | USGP | 80 | 3.42 | 0.544 | | | UNESUM<ESPAM |
| | ESPAM | 220 | 3.53 | 0.496 | | | |
| | UNESUM | 91 | 3.31 | 0.677 | | | |

Note: $p$ = significance (**$p < 0.01$); IMP = Importance; DEM = Demand; COM = Competence; UTM = Technical University of Manabi; ULEAM = Lay University Eloy Alfaro of Manabi; USGP = University San Gregorio of Portoviejo; UNESUM = State University of the South of Manabi; ESPAM = Agricultural Polytechnic School of Manabi.

knowledge and skill in the techniques of tutoring; inter- and intrapersonal competence; know-how and acceptance of criticism.

The three-factor model proposed for Q-AGT has confirmed the construct validity, showing an appropriate metric quality and a satisfactory fit of the model, and giving a scale with high reliability indices. In this way, a scale is obtained that is useful for generalizing results on guidance and tutoring in Latin American university students. This contributes to the assessment of various questions that have appeared as deficient, or little studied, in previous research, such as: a) attending and understanding the academic, personal and professional needs of the students through individualized support [14, 22, 29]; b) possible low skill in tutoring and university guidance in the teaching staff of Latin American universities [30]; and c) contribute to

**Table 6. Differences in the Q-AGT factors by university degree course.**

| Factor | University degree courses | | | | t student | |
|---|---|---|---|---|---|---|
| | S (n = 333) | | SC (n = 500) | | | |
| | M | SD | M | SD | t | d |
| **IMP** | 3.28 | 0.650 | 3.14 | 0.663 | -2.855** | 0.21 |
| **DEM** | 3.29 | 0.644 | 3.21 | 0.632 | 0.096 ns | 0.13 |
| **COM** | 3.41 | 0.593 | 3.41 | 0.553 | 0.999 ns | 0.00 |

Note

**$p < 0.01$; ns = not significant; d = Cohen's d; IMP = Importance; DEM = Demand; COM = Competence; S = Sciences; SC = Social Sciences.

evaluating a criterion of academic effectiveness related to admission, retention and support in the educational process of students [32]. In this sense, Q-AGT would be considered of great importance for initial courses, as an early detection mechanism for the prevention of academic dropout.

We agree with Ponce Ceballos, Aceves Villanueva, and Boroel Cervantes [54], who point out the importance of continuing to explore the evaluation of guidance and tutoring for the improvement of the student support processes and the training of teachers to undertake this activity in the university environment. This is related to superior performance and also greater persistence [8].

With regard to the application of Q-AGT, in the first factor -Importance-, the students attribute great value to university tutoring as an environment in which to promote personal development, especially autonomy, self-esteem and identity. In line with previous studies [1, 14, 16, 55], the importance of guidance and tutoring in the university context for personal, academic and professional development is underlined. These three types of development, which must be addressed by responding to the needs of students throughout their time at the university, support the idea of the progress of university tutoring towards a comprehensive model in which the personal dimension becomes especially important [15].

Looking at the second factor obtained -Demand-, the students express the need for guidance to help with job placement, as well as support and academic monitoring by the teaching staff of their university studies. These results are consistent with those obtained in other studies carried out in Europe and Latin America [9, 54], where students say that they feel a greater need for backup and support, specifically in important areas related to their future skills and employability −e.g. analysis of strengths and weaknesses, strategies for coping with stress and adversity, etc.−. This type of skill is in high demand from employers, as shown by the recent studies of González [56] and Bartual and Turmo [57].

Finally, with regard to the third factor −Teaching competence− the scale groups together the variables related to the competence of university teaching staff for tutorial work, such as intra- and interpersonal, technical, academic skills, and skills to help with job placement. Similarly, Pérez-Serrano et al. [58], highlight the importance of orientation and *"Face to face"* advice for university tutorials. Other authors, such as Pantoja-Vallejo et al. [13], in their study to validate a scale for measuring and assessing the guidance and tutorial practice, concluded that to the skills required for the tutorial work of the teaching staff should be added a cross-curricular competence, which comes from the domain of ICT.

Applying the scale has also made it possible to extract relevant, interesting information from the study. It has not been shown that women and men [20, 24] disagree on the assessment of university tutoring. Both sexes value highly the importance of tutoring in the universities of Ecuador, the demands that are met through it, and the competence of the teaching staff to carry it out.

The discrepancies between different universities suggest the strengths of university guidance and tutoring programs from the student's point of view. Although all the universities expressed a good, high valuation in this regard. Those students who are studying science [59, 60] value the importance of college tutoring more highly.

## Strengths, limitations, and future lines of research

One of the main limitations of this study is that it would have been desirable to expand the sample with subjects from other Latin American universities. In addition to university students, the opinion of other agents involved in university guidance and tutoring, such as teachers, would also be of interest.

Although the methodology, a cross-sectional approach using a survey, is a method commonly used in the study of the subject, it prevents the identification of causal relationships with other constructs. Also, the absence of studies carried out in Ecuador means that comparisons cannot be made that might contribute to deepening the scientific knowledge related to university tutoring and guidance.

Despite everything mentioned above, work is being done to overcome these limitations. First, data is currently being collected from a larger sample and the intention is to extend the assessment of counseling and mentoring to other Latin American countries. It is likewise intended to expand the group, in future studies, to university faculty, and to include the record of other variables related to the psychological well-being of students and teachers.

## Conclusions

The main conclusion of this study is that a valid and reliable instrument has been obtained to measure the performance of guidance and tutoring in Ecuadorian universities, which is theoretically grounded and operationally defined based on three factors: a) Importance; b) Demand; and c) Competence. The results of the Exploratory Factor Analysis show that the factors obtained and their loads correspond to the prior theoretical approaches.

The students have given a high assessment of the importance and demand of tutoring in the Ecuadorian university context, and of the teaching competence of the staff to meet the demands that satisfy thereby, all regardless of gender.

In relation to the participating universities, the student body of the Technical University of Manabi is the one that gives less importance to tutoring and to the demands generated by the student body. In contrast, the students of the Agricultural Polytechnic School of Manabi attach great importance to tutoring and believe that their teachers have good skills in tutoring. It is true, however, that despite the discrepancies, all universities scored highly in the assessment of tutoring and university guidance.

Considering the difference between the university degree courses in Ecuador, the results show that the students of degrees in Sciences are those who most value university tutoring.

Finally, it gives great importance to university orientation and tutoring as a strategy for the holistic development of the student -personal, academic and professional- and as a preventive against academic dropout. This study assesses whether this process is carried out properly, in line with other recent research aimed at the development of valid and reliable instruments [15, 16, 41]. Likewise, it is expected that this study will contribute to shed light on the discrepancy found between studies in terms of the factorial dimensions of tutoring, and provide knowledge in light of the very limited literature in Latin America, and specifically in Ecuador.

## Supporting information

**S1 Appendix. Cuestionario para la Evaluación de la Orientación y la Tutoría en Educación Superior (Q-AGT).**
(DOCX)

**S2 Appendix. Questionnaire for the Assessment of Guidance and Tutoring in Higher Education (Q-AGT).**
(DOCX)

## Author Contributions

**Conceptualization:** María Isabel Amor, Kasandra Vanessa Saldarriaga Villamil.

**Data curation:** Kasandra Vanessa Saldarriaga Villamil.

**Formal analysis:** Irene Dios.

**Investigation:** Kasandra Vanessa Saldarriaga Villamil.

**Methodology:** María Isabel Amor, Irene Dios.

**Supervision:** María Isabel Amor, Irene Dios.

**Validation:** Irene Dios.

**Writing – original draft:** María Isabel Amor, Kasandra Vanessa Saldarriaga Villamil, Irene Dios.

**Writing – review & editing:** María Isabel Amor, Irene Dios.

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
