## [Decision Letter · Decision Letter 0]

6 Apr 2021

PONE-D-21-02928

Assessing University Guidance and Tutoring in Higher Education: Validating a questionnaire on Ecuadorian students

PLOS ONE

Dear Dr. Irene Dios,

Thank you for submitting your manuscript to PLOS ONE. After careful consideration, we feel that it has merit but does not fully meet PLOS ONE’s publication criteria as it currently stands. Therefore, we invite you to submit a revised version of the manuscript that addresses the points raised during the review process.

The reviews are in general favourable and suggest that, subject to minor revisions, your paper could be suitable for publication. Please consider these suggestions, and I look forward to receiving your revision.

Although the theoretical framework is clear and well developed it would be good if it includes more references to international studies. The methods and techniques were well used to validate the questionnaire, however it would be good if you clarify why you started the likert scale with zero. Furthermore, you can explore and cross sociodemographic data with the 21 itens. Finally a minor remark: correct on  Page 16/28 - line 302, remove the word "express".

We look forward to receiving your revised manuscript.

Kind regards,

Teresa Carvalho

Academic Editor

PLOS ONE

Journal Requirements:

Please ensure that you include a title page within your main document. We do appreciate that you have a title page document uploaded as a separate file, however, as per our author guidelines (http://journals.plos.org/plosone/s/submission-guidelines#loc-title-page) we do require this to be part of the manuscript file itself and not uploaded separately.

Reviewers' comments:

Reviewer's Responses to Questions

**Comments to the Author**

1. Is the manuscript technically sound, and do the data support the conclusions?

Reviewer #1: Yes

Reviewer #2: Yes

2. Has the statistical analysis been performed appropriately and rigorously? 

Reviewer #1: Yes

Reviewer #2: I Don't Know

3. Have the authors made all data underlying the findings in their manuscript fully available?

Reviewer #1: Yes

Reviewer #2: Yes

4. Is the manuscript presented in an intelligible fashion and written in standard English?

Reviewer #1: Yes

Reviewer #2: Yes

5. Review Comments to the Author

Reviewer #1: The methods and techniques were well used to validate the questionnaire.

Regarding the Likert scale, why does it start at zero?

The Sociodemographic data could be better explored and crossed with the 21 items.

Most references are not in English but in Spanish and reference number 12 is not listed.

Reviewer #2: The study is well written and very interesting. Moreover, as highlighted by the authors, it aims at bridging a gap in the literature regarding this domain and they seem successful. They also point to ways of improving the study (e..g including more and diverse actors, and this indeed can be made in future work). Minor remarks: Please correct on Page 16/28 - line 302, remove the word "express".

6. PLOS authors have the option to publish the peer review history of their article (what does this mean?). If published, this will include your full peer review and any attached files.

Reviewer #1: **Yes: **Carolina Costa

Reviewer #2: No

---

## [Author Response · Author response to Decision Letter 0]

26 Apr 2021

Dear Reviewers,

Firstly, we would like to thank you for the suggestions made by the Academic Editor and the Reviewers. We have dealt with all the suggestions and they have been included in the manuscript. We thank the Reviewers for their indications, which have provided our research with better understanding and higher quality.

International studies have been included in the theoretical framework. In this regard, we have added two studies (7 and 8) which are considered of interest and in line with topic under consideration. These studies had not been included in the initial version of the manuscript because they have recently been published (April 2021). These publications are very recent and high impact, so we consider that the subject is original, novel and current.

Regarding the Reviewer's question 1 related to the Likert scale, the authors clarify that the scale has 5 response options with a graduation from “0” to “4”: where “0” is “completely disagree”, “1” is "disagree", "2" is "neutral", "3" is "agree", and "4" is "completely agree". A different graduation could have been indicated, for example from “1” to “5”, but it was decided to start with “0” considering the previous studies published by the authors (e.g., Dios et al., 2018. Validation of the Scale of Organizational and Didactic Competencies for Educators). We think that both options are acceptable and they do not determine the results obtained or the validation of the scale.

Many references are in Spanish since it is the language of the country where the scale has been validated. Although the review includes international studies in Spanish and English, the language of publication in Latin America, and specifically in Ecuador, is Spanish. Most literature about counseling and tutoring in this country is published in Spanish. However, as indicated by the Academic Editor and Reviewer 1, we have included international studies published in English in the new version of the manuscript.

In response to suggestions from Reviewer 1, the sociodemographic data have been analyzed and cross-referenced with the 21 items. It has been reflected in table 3 in the results section "Application of Q-AGT: assessment of university tutoring based on sex, university and university degree course". The reference number 12 has been included; “Authors, 2017” is substituted by “Amor MI, Dios I. Analysis of the psychometric

properties of a scale on the needs of students in Tutoring. Revista de Pedagogía. 2017; 38: 35-56 ". The reference 12 in the original manuscript is now the reference 14.

The errata indicated by Reviewer 2 has been corrected. The word "express" has been deleted.

Finally, we include a manuscript called “Revised Manuscript with Track Changes”, which contains the changes made to the original version and another version called “Manuscript” with the incorporated changes.

We have made sure to check that the manuscript complies with the PLOS ONE style requirements. Besides, the figures have been uploaded and downloaded to the Preflight Analysis and Conversion Engine (PACE). We provide figure in this format. We have also included the supporting information section in the final manuscript.

As it has been indicated by the Academic Editor, we have included the title page in the main document. We have checked that the reference list is complete and correct. The changes incorporated in the reference list have been three. We have included two references to international scientific studies, recommended by the Editor. Reference 14 (reference 12 in the original manuscript) which was incomplete (indicated by Reviewer 2) has been included.

We appreciate the contributions indicated.

Yours sincerely

---

## [Editor Report · Decision Letter 1]

4 Jun 2021

Assessing University Guidance and Tutoring in Higher Education: Validating a questionnaire on Ecuadorian students

PONE-D-21-02928R1

Dear Dr. Dios,

We’re pleased to inform you that your manuscript has been judged scientifically suitable for publication and will be formally accepted for publication once it meets all outstanding technical requirements.

Kind regards,

Andrew R. Dalby, PhD

Academic Editor

PLOS ONE
---

## [Editor Report · Acceptance letter]

7 Jun 2021

PONE-D-21-02928R1 

Assessing University Guidance and Tutoring in Higher Education: Validating a questionnaire on Ecuadorian students 

Dear Dr. Dios:

I'm pleased to inform you that your manuscript has been deemed suitable for publication in PLOS ONE. Congratulations! Your manuscript is now with our production department. 

Kind regards, 

on behalf of

Dr. Andrew R. Dalby 

Academic Editor

PLOS ONE